# Quality of Life of Physically Active and Inactive Women Who Are Older after Surgery for Stress Urinary Incontinence Using a Transobturator Tape (TOT)

**DOI:** 10.3390/jcm10204761

**Published:** 2021-10-17

**Authors:** Gabriela Kołodyńska, Maciej Zalewski, Felicja Fink-Lwow, Anna Mucha, Waldemar Andrzejewski

**Affiliations:** 1Department of Physiotherapy, Wroclaw University of Health and Sport Sciences, 51-612 Wrocław, Poland; felicitas1@wp.pl (F.F.-L.); waldemar.andrzejewski@awf.wroc.pl (W.A.); 2Faculty of Medical Sciences and Health Sciences, University of Social and Medical Sciences in Warsaw, 04-367 Warszawa, Poland; 3Department of Gynaecology and Obstetrics, Faculty of Health Sciences, Medical University of Wrocław, 50-367 Wrocław, Poland; zalewskim@interia.pl; 4Independent Public Health Care Center of the Ministry of the Interior and Administration in Wroclaw, Department of Gynaecology, 50-233 Wrocław, Poland; 5Department of Genetics, Wrocław University of Environmental and Life Sciences, 50-375 Wrocław, Poland; anna.mucha@upwr.edu.pl

**Keywords:** stress urinary incontinence, physical activity, quality of life

## Abstract

Urinary incontinence is a major health problem. According to various authors, it concerns 30–40% of the population and grows with age, affecting approximately 50% of women aged over 70. According to the recommendations of the International Continence Society, the treatment of urinary incontinence should commence with conservative treatment and, above all, with physiotherapy. If the conservative treatment fails or the level of urinary incontinence is too high, surgery is recommended. With regard to female patients examined at work, the TOT method was applied. The aim of this study was to assess the relationship between regular physical activity and the quality of life of women aged 65–87 who underwent surgical treatment for stress urinary incontinence (SUI) using the TOT method. The study group involved 60 postmenopausal women, patients of the Department of Gynaecology of the Hospital of Ministry of the Interior and Administration in Wroclaw, with SUI diagnosed during ultrasonography. The female patients were surveyed before and 12 months after the surgery using standardised IPAQ and WHOQOL-BREF.FL questionnaires. Significant positive relationships between quality of life and physical activity before and 12 months after the surgery were demonstrated in the somatic and social domains. Physically active postmenopausal women presented higher values in all domains and total quality of life according to the WHOQOL-BREF compared with physically inactive women, both before and 12 months after the procedure using the TOT method.

## 1. Introduction

The problem of urinary incontinence (UI) in the female population is extremely common [1]. According to various authors, it affects 30–40% of the population [2,3] and increases with age, affecting as much as 50% of women over 70 years of age [4]. This dysfunction contributes to a significant reduction in their quality of life. Since the 1970s, the literature has shown strong relationships between lifestyle and holistic health, and the prevention of chronic diseases. Research shows that regular physical activity and appropriate daily activity improve overall health [5,6,7,8]. Stress urinary incontinence (SUI) is the most common dysfunction associated with urinary incontinence, accounting for 50% of all cases of micturition [9,10]. SUI symptoms appear not only during physical effort but also when one coughs, laughs, or sneezes.

SUI is an uncontrolled leakage of urine caused by dysfunction of the bladder closure mechanism [11]. A special group in this context are postmenopausal women, where this dysfunction is observed in half of the population, with trends increasing with age. Urinary incontinence has also been found in young women engaged in professional sports [12,13]. This enables us to conclude that vigorous-intensity exercise may predispose to stress urinary incontinence [14,15]. However, most of the available studies on the relationship between regular physical activity and SUI concern young female athletes and focus less frequently on women exercising recreationally [16].

According to the recommendations of the European Association of Urology, the treatment of urinary incontinence should commence with conservative treatment and, above all, with physiotherapy [17]. Physical exercise, appropriately selected by a physiotherapist, leads to strengthening of the pelvic floor muscles in coordination with the abdominal muscles, thus causes the narrowing (urethral stricture) and elevation of the urethra, which prevents uncontrolled urinary leaks. If a conservative treatment fails or the level of urinary incontinence is too high, surgery is recommended [18]. One of the methods of surgical management—in terms of the high efficiency of treatment and low percentage of complications—is the treatment of SUI using midurethral tapes. The transobturator tape (TOT) procedure proposed in 2001 by Delorme [19] is particularly popular.

With regard to the female patients examined at work, the TOT method was used because it is as effective as the TVT method (implantation of a tension-free vaginal tape) but the time of the procedure is half as short and, for these reasons, preferred [20]. At the same time, according to the recommendations of the International Continence Society, physical activity leading to strengthening of the pelvic floor muscles facilitates the procedure and faster recovery after surgery. Undoubtedly, an improvement in women’s quality of life can be expected after effective therapy, but it seems that recreational, regular physical activity can be treated as an additional stimulator in improving their quality of life.

Stress urinary incontinence significantly reduces the quality of life in patients and affects both their physical and mental spheres [21]. Due to the symptoms, people with SUI are very often forced to limit their social roles and activities. They often withdraw from social and family life. An important problem is also the fact that patients are often ashamed to talk about their ailments to others. Many patients also live under the erroneous belief that SUI is a natural phenomenon, closely related to the aging process. The impact of incontinence on their QoL depends to a large extent on the severity of the symptoms.

In the case of significant ailments, their performance of basic life activities is disturbed. Hunskaar estimated that, in patients with severe symptoms of SUI, depressive symptoms occur in as many as 80% of respondents and that, with a low degree of ailment, the number of these women oscillates around 40% [22]. Irwin et al. have shown that patients with incontinence very often have a feeling of loss of control over their body, which additionally increases their psychological discomfort and reduces their life activity [23].

In the study by Milsom, he assessed SUI in the context of performing daily duties. Almost 100% of the women indicated that this problem reduces their quality of life. Over 50% of respondents stated that these ailments hinder their daily functioning. The researcher observed the occurrence of emotional disorders in 45% of patients. Moreover, he pointed out that urinary incontinence also has a direct impact on deteriorations in the quality of life of their families [24].

Sung et al. estimated the effect of a midurethral sling on improving leisurely physical activity levels and physical functioning in women with SUI. The women completed validated questionnaires for incontinence, leisure physical activity, and physical functioning before surgery and 6 months after surgery. The authors made a general conclusion. A midurethral sling and subsequent improvements in urinary incontinence are associated with improved leisure physical activity levels and physical functioning [25].

The authors of this article decided to take up this topic due to the small number of studies on the assessment of physical activity in patients after treatment with a midurethral sling and due to the lack of studies that investigate physical activity and quality of life. This topic is of significant clinical importance because it concerns an assessment of how the resultant surgery influenced the lives of patients.

The aim of the study was to assess the impact of the level of physical activity in patients after the TOT procedure and its impact on the quality of life of these women.

## 2. Material and Methods

### 2.1. Design and Data Collection

Sixty postmenopausal female patients, in whom menstruation had naturally subsided, were eligible for the project, and those who correctly completed the questionnaires both before and after the TOT operation were included in the study. This study is a preliminary report. The number of patients who qualified for the study depended on the number of TOT procedures performed by us and on certain inclusion and exclusion criteria. Those patients who stayed in the ward while waiting for the TOT procedure during the first phase of the project, who met the inclusion and exclusion criteria for the study, and who gave informed and written consent to participate in the study were qualified to participate in the project. Women aged 65–87 qualified for the project because it is after the age of 65 that SUI occurs in one in two women. The upper limit of the age range is due to the fact that TOT surgery is not usually performed in older patients.

The study group involved female patients of the Department of Gynaecology of the Hospital of Ministry of the Interior and Administration in Wroclaw who were diagnosed with SUI on the basis of ultrasound scan and anamnesis. Prior to the project, all female patients had obtained informed consent to participate in the study for the purposes of this paper. All selected female patients completed International Physical Activity Questionnaire (IPAQ)—Long Last Seven Days Self-Administrated and World Health Organisation Quality of Life (WHOQOL-BREF) questionnaires twice: on admission to the hospital and 12 months after the surgery. Questionnaires validated in the Polish language were used in the research. Patients were eligible for the research on the basis of specific inclusion and exclusion criteria. The project was approved by the Bioethics Committee of the Wroclaw Medical University with the number KB-806/2018.

Inclusion criteria:Type II/III stress urinary incontinence confirmed by ultrasound scan and medical history and confirmed by a gynaecologist;Not taking hormone replacement therapy (HRT) before and after surgery; andGave written informed consent to participate in the project.

Exclusion criteria:Overactive bladder;Mixed incontinence;Fistulae within the urinary tract;Congenital and acquired defects of the urethra or bladder;Urinary tract infections;The use of drugs that affect the overactive bladder; andTraining for sports professionally.

All of the female patients selected for the project were treated with the procedure of inserting the midurethral tapes using the transobturator (TOT) method [26]. In all patients, the procedure was performed vaginally. The principles during the surgery included the following procedures:Inserting a catheter in the urinary bladder;Incision and dissection of the vaginal mucosa and fascia;Proper insertion of the tape;Preventing the implants from wrapping and rolling up;Administering antibiotics; andOptimal tension-free stitching of the vaginal walls.

### 2.2. Measures

Standardised IPAQ and WHOQOL-BREF questionnaires were used in the study. The female patients completed the questionnaires twice, i.e., directly before the procedure and 12 months after the procedure.

#### 2.2.1. Assessment of Physical Activity

The level of physical activity was assessed using the International Physical Activity Questionnaire (IPAQ) Long Last Seven Days Self-Administrated Format [27,28]. The amount of physical activity was expressed in standard metabolic equivalent units (MET) as multiples of resting metabolic rate by minutes of performance during a week (MET-minutes/week) [29]. The total energy expenditure expressed in MET-minutes/week was a sum of individual energy expenditures during high-, moderate-, and low-intensity activities. Using the IPAQ categorical scoring, we distinguished two groups of subjects. Their total levels of physical activity were classified as active when Total ≥ 600 MET-min/week and inactive when Total < 600 MET-min/week. The level of MET min at which the cut-off between active and inactive was made was validated, in this publication, based on moderately to vigorously active [5].

#### 2.2.2. Assessment of Quality of Life

The quality of life of female patients was assessed using the World Health Organisation Quality of Life (WHOQOL-BREF) questionnaire. This questionnaire consists of 26 questions and is used to assess the quality of life of both people who are healthy or sick. The questions concern the following spheres of life: physical, mental, social, and environmental functioning. The results of the individual spheres are positive, which means that the higher the number of points, the more their quality of life increased [30,31].

All anthropometric characteristics, including body weight, body height, and body mass index (BMI), were measured and calculated according to guidelines from the literature [32].

### 2.3. Statistical Analysis

All women who were examined were divided into two groups based on their levels of physical activity. A woman with less than 150 min of moderate to vigorous physical activity per week was treated as physically inactive one. The level of MET-min at which the cut-off between active and inactive was made was validated, in this publication, based on moderately to vigorously active minutes in accordance with the methodology used in previous publications [33].

The basic descriptive statistics of the domain scores were determined using the *pastecs* package [34]. The statistical significance of the influence of surgery on the domain scores was verified with a Wilcoxon signed-rank test for dependent samples. The statistical significance of the influence of physical activity on the domain scores was verified with a Wilcoxon signed-rank test for independent samples. The statistical significance of the influence of surgery and physical activity on the domain scores was verified using a Kruskal–Wallis nonparametric analysis of variance. This method was used because of non-compliance of the trait distribution considered with normal distribution. After the Kruskal–Wallis test, multiple comparison tests were performed using the *agricolae* package [35].

The statistical significance of the impact of the analysed groups on the frequency of individual responses to the questions in the questionnaires considered was verified using Fisher’s exact test.

The relationship between the domain scores was investigated using Spearman’s rank correlation with the *psych* package [36]. The correlation in which 100% (1−α/k) confidence intervals did not overlap was considered statistically significantly different between groups [37].

The influence of the domain scores on the physical activity was considered with binary logistic regression. Physical activity or a lack thereof is a binary variable (active and inactive). In order to check whether a certain number of points awarded by patients after surgery within individual domains (domain scores) is related to undertaking physical activity, it was necessary to use this method. *p*-values below 0.05 indicate a statistically significant relationship between undertaking physical activity depending on the level of quality of life assessment. The OR value, if the probability of taking up physical activity increases with an increase in the rating (within the domain) by one point, i.e., OR = 1.0403, means that an increase in domain scores by one point results in an increase in the probability of physical activity by 4%.

## 3. Results

Sixty patients participated in the study. The mean age of the patients was 70.28 ± 5.19 years. The anthropometric data necessary to calculate BMI are shown in Table 1. The majority of postmenopausal women presented excess body weight (67%).

All patients eligible for the project had given birth, 92% (*n* = 55) of them by spontaneous labour and the others by Caesarean birth. The average number of births in the group was 2 ± 0.72, with the number of deliveries varying from 1–4. From the medical history conducted, 39 (65%) of the participants declared performing regular physical activity before the therapy. In the study group, no women practised competitive sports in the past, only recreational ones. Thirty-five percent of women were economically active while the others were pensioners. After the TOT treatment, all patients experienced a reduction in SUI symptoms.

The TOT treatment is characterised by low invasiveness and very high efficiency: according to studies conducted so far, it exceeds 90%, which is why, in our research we did not assess the effectiveness of the therapy itself but only its impact on physical activity and quality of life.

On the basis of the results obtained from the IPAQ questionnaire, the female patients were divided into two groups: physically active and inactive. The assessment of particular types of physical activity in the IPAQ questionnaire was based on the calculation of the intensity factor.

Both before and after the procedure, the study group was dominated by female patients with low levels of physical activity (44 vs. 36); however, 12 months after the procedure, the number of female patients presenting the recommended level of physical activity increased (24 vs. 16) (Table 2).

The results of the assessment of the quality of life are presented in the tables below (Table 3 and Table 4). In all of the domains studied (somatic, psychological, social, environmental, and total quality of life (TOTAL)) before and 12 months after the procedure, a statistically significant improvement was shown (Table 3). The highest values before and after the procedure were shown for the social domain. Regardless of the level of physical activity, the procedure performed had a positive impact on improving the quality of life of the female patients.

The total quality of life assessed in the WHOQOL-BREF questionnaire for both active and inactive women improved 12 months after surgery (Table 4). Statistically significant results were obtained for the somatic and total domains. Statistically significant differences between physically active and inactive women were demonstrated both before and after the surgery (*p* < 0.05). The results appear to indicate that, regardless of physical exercise status either before or after surgery, there has been an improvement in QoL in the somatic domain, which may be related to the improvement in SUI.

Twelve months after the procedure, physically active women presented the highest values in all domains and TOTAL. These values were statistically significant for somatic and environmental spheres compared with physically inactive women 12 months after the surgery. 

The observed changes are small but statistically significant. Changes of this magnitude are of clinical significance. The results obtained show that, not only immediately after the TOT procedure but also 12 months after the procedure, there was a significant improvement in the quality of life of the patients. After surgery, physically inactive women were able to become active and that active women were able to remain so, with gains in QoL.

Table 5, Table 6 and Table 7 present the correlations of the WHOQOL-BREF questionnaire results in all four domains in the whole study group (Table 5), before and after the surgery (Table 6), and takes into account the division between physically active and inactive female patients (Table 7). Table 8 shows the relationship between physical activity and the results obtained in the individual domains 12 months after the surgery.

The correlation results obtained before the procedure showed statistically significant relationships among individual domains. The results obtained indicate a statistically significant relationship between the environmental domain and somatic, psychological, and social domains. Statistically significant correlations also occurred between the social domain, and psychological and somatic domains. Furthermore, a statistically significant correlation was noted between the psychological and somatic domains (Table 5).

Moderate to high physical activity was associated with a significantly higher odds ratio of quality of life in the somatic domain (1.04 times, *p* < 0.0025) and the social domain (1.03 times, *p* < 0.0432) in postmenopausal women 12 month after the surgery (Table 8).

## 4. Discussion

The assessment of quality of life is significant for patients with stress urinary incontinence. It allows i.a. for an optimal choice of treatment methods and, consequently, has a positive impact on the outcome of the applied therapy. A significant deterioration in the quality of life often prompts female patients to see a doctor and to take up treatment. In some cases, subjectively experienced symptoms determine whether the female patient needs surgery or only conservative treatment [38].

Current publications confirm that physical activity has a positive impact on improving mental health, relieving stress, or creating positive perception of one’s health and self-esteem [39]. Positive relationships between physical activity and quality of life have been confirmed in many studies concerning people who are older, including patients with dementia, cardiovascular diseases, osteoporosis, or renal diseases [40].

The results of our research confirm that, especially in physically active patients, the overall quality of life improved after the TOT procedure. The highest total quality of life, perceived state of health, and quality of life in individual spheres—physical, psychological, social, and environmental—were observed in physically active female patients 12 months after surgery, i.e., in those with at least moderate physical activity (i.e., more than 600 MET-min/week). Similar results concerning physical activity and quality of life were shown by other authors. The studies conducted by Puciato et al. and Xu et al. have shown that physically active people perceive the individual components affecting the quality of life better than their physically inactive peers [41]. An improvement was observed in respondents in which the physical activity level was sufficiently high and consistent with the standards for duration and frequency per week [42].

Positive and statistically significant correlations between physical activity and the psychological domain of quality of life have been observed in the research by Chai et al. They have demonstrated the relationship between physical activity and optimism, joy of life, self-esteem, and depression [43]. On the other hand, Brown et al. found significant improvement in quality of life in subjects undertaking moderate physical activity for at least 30 min per day for 5 days a week or intensive physical activity for at least 20 min per day for 3 days a week [44]. In studies conducted in women aged 45–59, Guimarães and Baptista noted that undertaking moderate physical activity for 60 min per day was positive and significantly correlated with women’s quality of life [45]. The results of our observations also confirm the positive significance of regular physical activity in the group of women with SUI. Women who were considered physically active after the TOT procedure showed higher improvements in their quality of life compared with inactive women, in all domains.

Our results are clinically relevant as they clearly show that the physical activity of women after TOT surgery increased. This fact, in turn, may contribute to an improvement in the assessed quality of life. Along with the increase in the levels of physical activity in the studied patients, their lifestyles improved, which could have contributed to the reduction in the risk of chronic diseases related to a lack of exercise. Moreover, as shown by the abovementioned research results, when practicing sports, the well-being and mood of exercising people improve, which in turn causes an increase in the assessed parameters of the quality of life. The highest changes were seen in physically active women after surgery who were not physically active before surgery. This may be due to the fact that, as a result of the TOT procedures, they could begin to engage in physical activity without fear, which significantly changed their lives. Often, patients who suffer from SUI do not undertake any physical activity. They give up on it because symptoms of SUI often appear even when walking or running slowly. After the TOT procedure is performed and the recommendations are provided by the doctor, many women begin physical activity and thus change their lives, which, as shown in this research, also improves the quality of life.

The relationship between physical activity and the risk of stress urinary incontinence is not inconsistent. Despite numerous studies on these relationships, there are no clear results [46]. The available studies have an insufficient number of participants, are cross-sectional research, or use only self-designed questionnaires [47]. A description of the pathophysiology of stress urinary incontinence and its relationship to physical activity is an important element in terms of prevention and choice of treatment method [48]. Nevertheless, our studies need to be continued on a larger group of women.

## 5. Limitations

Our paper has certain limitations. The results obtained by the authors were based on an insufficient number of women included in the study. However, this was due to the limitations used to obtain a homogeneous group of postmenopausal women in terms of fertility and non-use of hormone replacement therapy. Moreover, there is no control group in the study, which may also have a negative impact on the conclusions drawn. To our knowledge, this is the first paper analysing the relationships between physical activity and quality of life in postmenopausal women before and 12 months after surgery.

## 6. Conclusions

The TOT procedure increased physical activity in the majority of patients.After the surgery using the TOT method, the quality of life improved in the somatic and social domains in women who are older regarding the assessment of 12 months after the surgery.After the surgery using the TOT method, physically inactive women were able to become active and active women were able to remain so, with gains in QoL.The WHOQOL-BREF questionnaire may be useful in clinical practice to assess the quality of life of female patients with SUI who underwent surgical treatment.

## Figures and Tables

**Table 1 jcm-10-04761-t001:** Characteristics of the studied group.

Characteristics	Mean ± SD	Median (Min–Max)
Age (years)	70.28 ± 5.19	69 (65–87)
Weight (kg)	69.94 ± 11.65	69.5 (48–100)
Body height (cm)	161.33 ± 5.40	162 (149–172)
BMI (kg/m^2^)	27.07 ± 4.69	26.74 (17.63–38.86)
Underweight, *n* (%)	1 (1.67%)	
Normal (healthy weight), *n* (%)	19 (31.67%)	
Overweight, *n* (%)	24 (40%)	
Obesity grade 1, *n* (%)	11 (18.33%)	
Obesity grade 2, *n* (%)	5 (8.33%)	
Menstruation has disappeared naturally	60 (100%)	

**Table 2 jcm-10-04761-t002:** Domain scores of the physically inactive and physically active women before and after surgery.

	Before Surgery *n* = 60	After Surgery *n* = 60
inactive women	44	36
MET	<600 MET-min/week	<600 MET-min/week
active women	16	24
MET	≥600 MET-min/week	≥600 MET-min/week

The data are presented as the number of patients.

**Table 3 jcm-10-04761-t003:** Domain scores of the WHOQOL-BREF questionnaire from women before and after surgery.

	Before Surgery *n* = 60	After Surgery *n* = 60	Effect Size	*p*-Value
somatic	42.86 (0–80) 44.14 (16.52)	51.43 (0–80) 50.86 (17.38)	0.500	0.00009
psychological	50.00 (0–73.33) 46.17 (16.43)	53.33 (0–76.67) 51.33 (16.27)	0.456	0.00016
social	53.33 (0–80) 50.44 (15.64)	56.67 (20–80) 54.11 (15.82)	0.313	0.00883
environmental	38.75 (0–60) 37.96 (12.84)	42.50 (20–67.50) 42.62 (13.97)	0.365	0.00098
TOTAL	40.52 (0–67.24) 39.70 (14.01)	45.69 (0–68.97) 45.14 (14.71)	0.496	0.00005

The data are presented as the median (range) and the mean (standard deviation). The given *p*-values are for the Wilcoxon test for dependent samples.

**Table 4 jcm-10-04761-t004:** Domain scores of the WHOQOL-BREF questionnaire from physically inactive and physically active women before and after surgery.

	Inactive Women before Surgery *n* = 44	Active Women before Surgery *n* = 16	Inactive Women after Surgery *n* = 36	Active Women after Surgery *n* = 24
somatic	40.00 (0–80) 41.62 ^a^ (16.83)	48.57 (22.86–71.43) 51.07 ^a,b^ (13.84)	48.57 (0–77.14) 46.98 ^a,b^ (17.53)	58.57 (28.57–80) 56.67 ^b^ (15.78)
psychological	48.33 (0–73.33) 44.39 (17.91)	51.67 (30–66.67) 51.04 (10.38)	53.33 (0–76.67) 50.19 (18.46)	53.33 (26.67–73.33) 53.06 (12.47)
social	53.33 (0–73.33) 48.64 (16.43)	60.00 (33.33–80) 55.42 (12.53)	53.33 (0–80) 52.04 (17.08)	60.00 (20–73.33) 57.22 (13.47)
environmental	37.50 (0–60) 36.36 (13.90)	42.50 (27.50–60) 42.34 (8.19)	41.25 (0–67.50) 41.25 (14.66)	43.75 (25–65) 44.69 (12.90)
TOTAL	38.36 ^a^ (0–67.24) 37.70 (14.80)	46.55 ^a,b^ (25.86–62.93) 45.20 (9.97)	45.26 ^a,b^ (0–68.10) 42.94 (15.70)	47.84 ^b^ (27.59–68.97) 48.46 (12.69)

The data are presented as the median (range) and the mean (standard deviation). Mean values differing statistically significant between the groups were marked by different letters: ^a^ inactive women, ^b^ active women (Kruskal–Wallis test, *p*-value < 0.05).

**Table 5 jcm-10-04761-t005:** Spearman’s correlation matrix of domain scores for the full data set before surgery.

	Somatic	Psychological	Social	Environmental
somatic	1.00	0.80 *	0.46 *	0.69 *
psychological		1.00	0.57 *	0.78 *
social			1.00	0.61 *
environmental				1.00

Coefficients marked with * were statistically significant (*p*-value < 0.05).

**Table 6 jcm-10-04761-t006:** Spearman’s correlation matrix of domain scores for the women before surgery (upper triangle) and for the women after surgery (lower triangle).

	Somatic	Psychological	Social	Environmental
somatic	1.00	0.81 *	0.46 *	0.69 *
psychological	0.78 *	1.00	0.52 *	0.73 *
social	0.44 *	0.61 *	1.00	0.54 *
environmental	0.67 *	0.82 *	0.66 *	1.00

Coefficients marked with * were statistically significant (*p*-value < 0.05). Corresponding correlation coefficients were not statistically significantly different (*p*-value > 0.05).

**Table 7 jcm-10-04761-t007:** Spearman’s correlation matrix of domain scores for the inactive women (upper triangle) and for the active women (lower triangle).

	Somatic	Psychological	Social	Environmental
somatic	1.00	0.79 *	0.36 *	0.65 *
psychological	0.86 *	1.00	0.57 *	0.78 *
social	0.59 *	0.57 *	1.00	0.57 *
environmental	0.76 *	0.78 *	0.67 *	1.00

Coefficients marked with * were statistically significant (*p*-value < 0.05). Corresponding correlation coefficients were not statistically significantly different (*p*-value > 0.05).

**Table 8 jcm-10-04761-t008:** The association between physical activity and score domains after surgery.

Domain	*p*-Value	OR	CI
somatic	0.0025	1.0403	(1.0086; 1.0774)
psychological	0.1037	1.0213	(0.9904; 1.0576)
social	0.0432	1.0308	(0.9960; 1.0746)
environmental	0.0513	1.0309	(0.9931; 1.0745)

The *p*-values are for the single logistic regression model. OR = odds ratio, CI = 98.75% confidence interval. The odds ratios were not statistically significantly different between domains.

## Data Availability

The datasets used and/or analysed during the current study are available from the corresponding author upon reasonable request.

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
