# Peer review of "Quality of Life of Physically Active and Inactive Women Who Are Older after Surgery for Stress Urinary Incontinence Using a Transobturator Tape (TOT)"

_jcm, 2021, doi:10.3390/jcm10204761_

Round 1

Reviewer 1 Report

Quality of life of physically active and inactive elderly women following surgery for stress urinary incontinence using a transobturator tape

the aim of this study was to assess the relationship between physical activity and quality of life for women who were underwent surgical treatment for stress urinary incontinence using transobturator tape using a pre post design

the paper needs considerable editing for clarity and brevity

the abbreviations used in the abstract are not defined but this would lead to excessive lengthening of the abstract perhaps the abbreviations can be ignored here?

The significance of the results is not mentioned in the abstract and diminishing its utility for the casual reader

Introduction;

the first two sentences can be edited for brevity many authors will not know what the somatic sphere is and some simplification of the explanation may be required

Do the authors mean gender or sex in terms of age-appropriate physical activity? You lie the introduction is somewhat longwinded, might be made more concise and less conversational in tone

Reference 17 is not ICS but EAU

The meaning of “efficient” therapy is unclear

The research question and its relative importance is lost in the general discussion.

This should be revised

Methods

How was the sample size determined – upon what basis?

Table 1 and discussion of the demographics should be a result, not as part of the method

IPAQ and WGOQOL – BREF need explaining in first use

the term bladder hyperactivity isn’t recognized in ICS terminology

is the level of MET- min at which the active and inactive cut was made validated? there also appear to be 2 definitions – one based on MET and the other on moderate to vigorously active minutes

the results appear to indicate that, regardless of physical exercise status either before or after surgery. QoL in the somatic domain improved, presumably related to the improvement in SUI?

Was there a difference in change in QoL between the inactive women who became active and active groups versus those who remained inactive?

What proportion of women had improved SUI?

The results section, along with the narrative, is impenetrable, making it difficult to understand exactly what the results show.  What validity is there in separating out the domains of the QoL score? The observed changes are small but statistically significant – what is the clinical significance of changes of this magnitude?

The main impact appears to be that physically inactive women were able to become active and that active women were able to remain so, with attendant QoL gain – but this would have been affected by the surgery outcome

Discussion

The authors should summarize their findings succinctly and contrast these two that which is known the confounding effects of the surgery and baseline physical activity levels need to be acknowledged. The influence of increased physical activityon quality of life following surgery is a major factor

The fact that the highest changes were seen in physically active women after surgery needs to be clarified in terms of those who changed their physical activity versus those who were able to remain active

Were there any women in this small sample who were highly active and may have fallen into the group with increased levels of stress urinary incontinence due to their athletic abilities

The authors need to discuss the significance of their results particularly in terms of the magnitude of change related to the clinical significance they also need to acknowledge the inherent weakness of their experimental design, the small sample and the lack of any control group.

Author Response

Dear Reviewer,

Enclosed herein I am Submitting the revised manuscr ipt Ref. No.: Jcm-1383347 . The text has been changed according to reviewers ' suggestions . Changes in the manuscript are marked .                          

Thank you very much for your thorough evaluation of my manuscript , valuable comments and help in improving.               

With kind regards ,   

Gabriela Kołodyńska

Reviewer # 1 : 

the paper needs considerable editing for clarity and brevity

Answer 

As recommended by the reviewers, we edited the entire article to be clarity and brevity

Reviewer # 1 : 

the abbreviations used in the abstract are not defined but this would lead to excessive lengthening of the abstract perhaps the abbreviations can be ignored here?

The significance of the results is not mentioned in the abstract and diminishing its utility for the casual reader

Answer

The definition of the abbreviation has been removed from the abstract. They were defined in the text on first use. The most important conclusions were also added to the summary

Introduction;

Reviewer # 1 : 

the first two sentences can be edited for brevity many authors will not know what the somatic sphere is and some simplification of the explanation may be required

Answer

We have revised this section of the introduction as recommended by the reviewer.

Reviewer # 1 : 

Do the authors mean gender or sex in terms of age-appropriate physical activity? You lie the introduction is somewhat longwinded, might be made more concise and less conversational in tone

Answer:

As recommended by the reviewer, we have made changes to the introduction. 

Reviewer # 1 : 

Reference 17 is not ICS but EAU

Answer:

Thank you. We changed it.

Reviewer # 1 : 

The meaning of "efficient" therapy is unclear

Answer:

Thank you. Should be "effective"

Reviewer # 1 : 

The research question and its relative importance is lost in the general discussion.

This should be revised

Answer:

We have made changes to the general discussion so that it is more related to the research question

Methods

Reviewer # 1 : 

How was the sample size determined - upon what basis?

Answer:

We have supplemented the description of the methods: 

“Sixty postmenopausal female patients, in whom menstruation had naturally subsided, were eligible for the project and which correctly completed the questionnaires both before and after the TOT operation. This study is preliminary report. The number of patients qualified for the study depended on the number of TOT procedures performed by us and the fulfillment of certain inclusion and exclusion criteria "

Reviewer # 1 : 

Table 1 and discussion of the demographics should be a result, not as part of the method

Answer:

We moved the table and its description to the results section

Reviewer # 1 : 

IPAQ and WGOQOL - BREF need explaining in first use

Answer:

Thank you. We added an explanation when using these abbreviations for the first time.

Reviewer # 1 : 

the term bladder hyperactivity isn't recognized in ICS terminology

Answer:

Thank you. We changed that concept.

Reviewer # 1 : 

is the level of MET- min at which the active and inactive cut was made validated? there also appear to be 2 definitions - one based on MET and the other on moderate to vigorously active minutes

Answer:

The level of MET- min at which the active and inactive cut was made validated. In this publication based on moderate to vigorously active minutes in accordance with the methodology used in publications Haskell (2007) and Porto (2012).

Haskell, WL, Lee, IM, Pate, RR, Powell, KE, Blair, SN, Franklin, BA, ... Bauman, A. (2007). Physical activity and public health: Up- dated recommendation for adults from the American College of Sports Medicine and the American Heart Association. Circulation, 116 (9), 1081-1093.

Porto, DB, Guedes, DP, Fernandes, RA, & Reichert, FF (2012). Perceived quality of life and physical activity in Brazilian older adults. Motricidade, 8 (1), 33–41.

Reviewer # 1 : 

the results appear to indicate that, regardless of physical exercise status either before or after surgery. QoL in the somatic domain improved, presumably related to the improvement in SUI?

Answer:

As suggested by the reviewer, in the section the results were added:

The results appear to indicate that, regardless of physical exercise status either before or after surgery there has been an improvement in QoL in the somatic domain, which may be related to the improvement in SUI.

Reviewer # 1 : 

Was there a difference in change in QoL between the inactive women who became active and active groups versus those who remained inactive?

Answer:

Before the surgery, 10 physically inactive women changed the mode to active after the surgery. Two women from the physically active group switched to the physically inactive group after the procedure. Due to such a small number, we did not perform statistical calculations for this change.

Reviewer # 1 : 

What proportion of women had improved SUI?

Answer:

We have added in the material and methods section:

“After the TOT treatment, all patients experienced a reduction in SUI symptoms. The TOT treatment is characterized by low invasiveness and very high efficiency: according to the studies conducted so far, it exceeds 90%, which is why in our research we did not assess the effectiveness of the therapy itself, but only its impact on physical activity and quality of life. "

Reviewer # 1 : 

The results section, along with the narrative, is impenetrable, making it difficult to understand exactly what the results show. What validity is there in separating out the domains of the QoL score? The observed changes are small but statistically significant - what is the clinical significance of changes of this magnitude?  

Answer:

We added the results in the section:

“The observed changes are small but statistically significant. The changes of this magnitude is the clinical significance. The obtained results show that not only immediately after the TOT procedure, but also 12 months after the procedure, there was a significant improvement in the quality of life of the patients. "

Reviewer # 1 : 

The main impact appears to be that physically inactive women were able to become active and that active women were able to remain so, with attendant QoL gain - but this would have been affected by the surgery outcome

Answer:

Yes, we agree with this conclusion. We added it in the results section and conclusions

Discussion

Reviewer # 1 : 

The authors should summarize their findings succinctly and contrast these two that is known the confounding effects of the surgery and baseline physical activity levels need to be acknowledged. The influence of increased physical activityon quality of life following surgery is a major factor

The fact that the highest changes were seen in physically active women after surgery needs to be clarified in terms of those who changed their physical activity versus those who were able to remain active

Answer:

We changed the discussion according to the reviewers' instructions.

Reviewer # 1 : 

Were there any women in this small sample who were highly active and may have fallen into the group with increased levels of stress urinary incontinence due to their athletic abilities

Answer:

We did not qualify people with increased levels of stress urinary incontinence due to their athletic abilities for our research, as this could distort our results. In fact, the paragraphs on SUI in women who play sports were not necessarily included in the discussion and could be confusing to the reader. We removed them. Additionally, in the Material and methods section, we added in the exclusion criteria: "Women training sports professionally."

Reviewer # 1 : 

The authors need to discuss the significance of their results particularly in terms of the magnitude of change related to the clinical significance they also need to acknowledge the inherent weakness of their experimental design, the small sample and the lack of any control group.

Answer:

We changed the discussion according to the reviewers' instructions. Additionally, in the Limitations section, we added:

“Moreover, there is no control group in the study, which may also have a negative impact on the conclusions drawn.”

Reviewer 2 Report

This study's conclusion is not supported by other published studies and contradicts the results of other studies. There are numerous English language grammatical errors. This type of questionnaire-based study has a very low impact and hence is not suitable for JCM. 

Author Response

Dear reviewer, Thank you very much for reading our manuscript and for your feedback. We would like to inform you that thanks to the suggestions and comments of other reviewers, we introduced many changes to our work, which significantly improved its quality. Kind regards,
Authors

Reviewer 3 Report

Overall comments: 

  1. The specific study question is unclear making it difficult to interpret analysis plan. What does “relationship between physical activity and quality of life” mean specifically in the context of women receiving the sling?
  2. Overall the methodology is not clear, starting from exactly what study design this is. Also need to better justify the study population (why only age 65-87?). There also needs to be a sample size justification/power calculation if this was prospectively performed.
  3. Would have been more useful to create 2 comparison groups (active vs non-active) that remained the same throughout time.
  4. The analytic techniques used are unclear in exactly what question they are designed to answer. For example, in the logistic regression analysis, what exactly is the scientific question that the authors are asking, and why is this question important? Also unclear what exploring the correlations between the domains of the survey adds to the overall study question.

Introduction

  • Needs to better establish existing literature to provide basis for the present study. For example, explanations regarding the etiology of SUI is not relevant to this study question (Page 2 paragraph 1).
  • Include more substantive information for example of why TOT may be associated with improved physical activity and quality of life, such as literature supporting impact of slings on physical function.

Materials and Methods

  • Why the specific age group decided upon of 65 to 87?
  • What time period was determined studied? Name type of study used in Design. Was this a prospective cohort study?
  • Why only natural menopause included? Why not surgical menopause as well? How was that distinction determined?
  • Some of the Results are in the Methods. For example, section 2.1 should be in Results.
  • Methods really starts on page 3. For measures used, specify whether they were used in Polish and whether they have been validated in the language.
  • Rather than anamnesis use “medical history”
  • Inclusion/exclusion criteria: Specify what system was used for Type II/III SUI. Why did this have to be confirmed by US scan? Why couldn’t women be taking HRT before/after surgery? Was this only systemic or vaginal or both?
  • For surgical technique, I assume ‘avoiding infection of implants’ means antibiotics. May not even need this section if authors used standard of care.
  • 2 – include if the measures were validated in Polish. Were anthoprometric measures abstracted from the medical chart or obtained specifically for the study?
  • 3 – Don’t need to specify R packages used.
  • 3 – Why was a woman with <150 minutes MVPA/week physically inactive?

Results

  • For a prospective study, what time period did enrollment take place and how many participants were approached, and how many were enrolled?
  • While the QOL assessments presented in Table 3 are all statistically significant, is this clinically significant? What is the MID for each of these measures? For example, the median psychological scores increased only from 50 to 53. Is that clinically significant?
  • In Table 4, the ‘n’ changing before vs after surgery is very confusing. And if the populations being compared are different (inactive women transitioning into active group after surgery) it is unclear how to interpret these comparisons. May be more clear to have 1 column of ‘preoperatively inactive’ and another of ‘preoperatively active.’
  • For Table 5-7: Also unclear what exploring the correlations between the domains adds to the overall study question. Further, the actual Spearman coefficient is what would be used to make the inference on the strength of the correlation, rather than only looking to a P-value of statistical significance.
  • For the logistic regression analysis (Table 8), what exactly is the scientific question that the authors are asking, and why is this question important? ‘The way the analysis is structured it seems: is each domain associated with the outcome of physical activity?’ – which is difficult to conceptualize. Further, interpretation of strengths of association based solely on significant P-values is incorrect.

Discussion

  • “The results of our research confirmed a significant correlation between the quality of life and the level of physical activity in the somatic and social spheres in female patients both before and after the SUI surgery with the use of TOT method.” Unclear how authors came to this conclusion based on the presented data.
  • Extraneous information on general stress incontinence mechanisms is presented in the last 3 paragraphs of page 7.
  • Page 8 paragraph 1: Unclear as to how this study refutes the “hammock hypothesis” – the fact that patients are able to be physically active due to treated SUI suggests evidence to support the “hammock hypothesis” as this suggests the sling (which is thought to work based on the hammock hypothesis) is actually working.
  • The present data do not support conclusions 1-3.

Author Response

Dear Reviewer,

Enclosed herein I am Submitting the revised manuscr ipt Ref. No.: Jcm-1383347 . The text has been changed according to reviewers ' suggestions . Changes in the manuscript are marked .                          

Thank you very much for your thorough evaluation of my manuscript , valuable comments and help in improving.               

With kind regards ,   

Gabriela Kołodyńska

Reviewer # 2:

Overall comments: 

  1. The specific study question is unclear making it difficult to interpret analysis plan. What does “relationship between physical activity and quality of life” mean specifically in the context of women receiving the sling?
  2. Overall the methodology is not clear, starting from exactly what study design this is. Also need to better justify the study population (why only age 65-87?). There also needs to be a sample size justification / power calculation if this was prospectively performed.
  3. Would have been more useful to create 2 comparison groups (active vs non-active) that remained the same throughout time.
  4. The analytic techniques used are unclear in exactly what question they are designed to answer. For example, in the logistic regression analysis, what exactly is the scientific question that the authors are asking, and why is this question important? Also unclear what exploring the correlations between the domains of the survey adds to the overall study question.

Answer:

  1. We clarified the purpose of our study: 

"The aim of the study was to assess the impact of the level of physical activity in patients after the TOT procedure and its impact on the quality of life of these women". 

  1. We changed the purpose of our research and refined the methodology. This research was not done prospectively. We did not plan the sample size and power of the test before starting the research. In the material and methods section, we added:

“Those patients who, during the first phase of the project, stayed in the ward while waiting for the TOT procedure, met the inclusion and exclusion criteria for the study, and gave informed and written consent to participate in the study, were qualified to the project. Women aged 65-87 were qualified for the project, because it is after the age of 65 that SUI occurs in every second woman. The upper limit of the age range is due to the fact that TOT surgery is not usually performed in older patients. "

  1. Of course, we could make such a calculation, but it would reduce the size of the groups. In addition, our goal was actually to show that after the TOT procedure, the physical activity of our patients increases, and thus also their quality of life improves.
  2. We have expanded the statistical analysis section:

“Physical activity or lack thereof is a binary variable (active, inactive). In order to check whether a certain number of points awarded by patients after surgery within individual domains (domain scores) is related to undertaking physical activity, it was necessary to use this method. p-values ​​below 0.05 indicate a statistically significant relationship between undertaking physical activity depending on the level of quality of life assessment. The OR value, if the probability of taking up physical activity increases with an increase in the rating (within the domain) by one point, ie OR = 1.0403 means that an increase in domain scores by one point results in an increase in the probability of physical activity by 4%. "

Reviewer # 2:

Introduction

  • Needs to better establish existing literature to provide basis for the present study. For example, explanations regarding the etiology of SUI is not relevant to this study question (Page 2 paragraph 1).
  • Include more substantive information for example of why TOT may be associated with improved physical activity and quality of life, such as literature supporting impact of slings on physical function.

Answer:

  • We have removed some unnecessary passages from the introduction
  • In the introduction, we wrote:

" Stress urinary incontinence significantly reduces the quality of life in patients and affects both the physical and mental spheres [21]. Due to the symptoms, people with SUI are very often forced to limit their social roles and activities. They often withdraw from social and family life. An important problem is also the fact that patients are often ashamed to talk about their ailments to others. Many patients also live under the erroneous belief that SUI is a natural phenomenon, closely related to the aging process. The impact of incontinence on the QoL depends to a large extent on the severity of symptoms.

In the case of significant ailments, the performance of basic life activities is disturbed. Hunskaar estimated that in patients with severe symptoms of SUI, depressive symptoms occur in as many as 80% of respondents, and with a low degree of ailment, the number of these women oscillates around 40% [22]. Irwin et al. have shown that incontinence patients very often have a feeling of losing control over their body, which additionally increases their psychological discomfort and reduces their life activity [23].

In his study, Milsom assessed SUI in the context of performing daily duties. Almost 100% of the women indicated that this problem reduces their quality of life. Over 50% of respondents stated that these ailments hinder their daily functioning. The researcher observed the occurrence of emotional disorders in 45% of patients. Moreover, he pointed out that urinary incontinence also has a direct impact on the deterioration of the quality of life of their families [24].

 Sung et al. to estimate the effect of the midurethral sling on improving leisure physical activity levels and physical functioning in women with SUI. Women completed validated questionnaires for incontinence, leisure physical activity, and physical functioning before surgery and 6 months after surgery. The authors made a general conclusion. Midurethral sling and subsequent improvements in urinary incontinence are associated with improved leisure physical activity levels and physical functioning [25].

The authors of this article decided to take up its topic due to the small number of studies on the assessment of physical activity in patients after midurethral sling and the lack of studies that would investigate physical activity and quality of life. This topic is of significant clinical importance because it concerns the assessment of how the resulting surgery influenced the lives of patients. "

Reviewer # 2:

Materials and Methods

  • Why the specific age group decided upon of 65 to 87?
  • What time period was determined studied? Name type of study used in Design. Was this a prospective cohort study?
  • Why only natural menopause included? Why not surgical menopause as well? How was that distinction determined?
  • Some of the Results are in the Methods. For example, section 2.1 should be in Results.
  • Methods really starts on page 3. For measures used, specify whether they were used in Polish and whether they have been validated in the language.
  • Rather than anamnesis use "medical history"  
  • Inclusion / exclusion criteria: Specify what system was used for Type II / III SUI. Why did this have to be confirmed by US scan? Why couldn't women be taking HRT before / after surgery? Was this only systemic or vaginal or both?
  • For surgical technique, I assume 'avoiding infection of implants' means antibiotics. May not even need this section if authors used standard of care.
  • 2 - include if the measures were validated in Polish. Were anthoprometric measures abstracted from the medical chart or specifically obtained for the study?
  • 3 - Don't need to specify R packages used.
  • 3 - Why was a woman with <150 minutes MVPA / week physically inactive?

Answer:

  • In the material and methods section, we added:

“Those patients who, during the first phase of the project, stayed in the ward while waiting for the TOT procedure, met the inclusion and exclusion criteria for the study, and gave informed and written consent to participate in the study, were qualified to the project. Women aged 65-87 were qualified for the project, because it is after the age of 65 that SUI occurs in every second woman. The upper limit of the age range is due to the fact that TOT surgery is not usually performed in older patients. "

  • This research was not done prospectively. We did not plan the sample size and power of the test before starting the research. In the material and methods section, we added:

“Those patients who, during the first phase of the project, stayed in the ward while waiting for the TOT procedure, met the inclusion and exclusion criteria for the study, and gave informed and written consent to participate in the study, were qualified to the project. Women aged 65-87 were qualified for the project, because it is after the age of 65 that SUI occurs in every second woman. The upper limit of the age range is due to the fact that TOT surgery is not usually performed in older patients. "

  • Only natural menopause patients were included in our study because we did not want other factors influencing the end of menopause to affect our results. Patients will be asked about it during the interview, and we have obtained this data from their treatment history.
  • We have moved the indicated tables and their description to the results section  
  • We moved some of the text that should not be included in the material and methods section to the results section. In the material and methods section, we added the following information: "The questionnaires were validated in the Polish language were used in the research."
  • We changed anamnesis to medical history.
  • The degree of urinary incontinence was determined by a gynecologist based on a physical examination. Additionally, in order to confirm the degree of dysfunction, doctors also use ultrasound. Patients who applied HRT were not eligible for the study, so that it would not affect their QoL assessment. In all patients, the procedure was performed vaginally.
  • Yes, when we say " avoiding infection of implants" we meant "antibiotics". We briefly described the procedure so that the reader would have no doubts as to what it looks like.
  • In the material and methods section, we added the following information: "The questionnaires were validated in the Polish language were used in the research." Anthropometric measurements were made specifically for this study.
  • We removed the information about R packages
  • The level of MET- min at which the active and inactive cut was made validated. In this publication based on moderate to vigorously active minutes in accordance with the methodology used in publications Haskell (2007) and Porto (2012).

Haskell, WL, Lee, IM, Pate, RR, Powell, KE, Blair, SN, Franklin, BA, ... Bauman, A. (2007). Physical activity and public health: Up- dated recommendation for adults from the American College of Sports Medicine and the American Heart Association. Circulation, 116 (9), 1081-1093.

Porto, DB, Guedes, DP, Fernandes, RA, & Reichert, FF (2012). Perceived quality of life and physical activity in Brazilian older adults. Motricidade, 8 (1), 33–41.

Reviewer # 2:

Results

  • For a prospective study, what time period did enrollment take place and how many participants were approached, and how many were enrolled?
  • While the QOL assessments presented in Table 3 are all statistically significant, is this clinically significant? What is the MID for each of these measures? For example, the median psychological scores increased only from 50 to 53. Is that clinically significant?
  • In Table 4, the 'n' changing before vs after surgery is very confusing. And if the populations being compared are different (inactive women transitioning into active group after surgery) it is unclear how to interpret these comparisons. May be more clear to have 1 column of 'preoperatively inactive' and another of 'preoperatively active.'
  • For Table 5-7: Also unclear what exploring the correlations between the domains adds to the overall study question. Further, the actual Spearman coefficient is what would be used to make the inference on the strength of the correlation, rather than only looking to a P-value of statistical significance.
  • For the logistic regression analysis (Table 8), what exactly is the scientific question that the authors are asking, and why is this question important? 'The way the analysis is structured it seems: is each domain associated with the outcome of physical activity?' - which is difficult to conceptualize. Further, interpretation of strengths of association based solely on significant P-values ​​is incorrect.

Answer:

  • This research was not done prospectively. We did not plan the sample size and power of the test before starting the research. In the material and methods section, we added:

“Those patients who, during the first phase of the project, stayed in the ward while waiting for the TOT procedure, met the inclusion and exclusion criteria for the study, and gave informed and written consent to participate in the study, were qualified to the project. "

  • We have added information about the clinical relevance of this study in several places in the article.
  • Before the surgery, 10 physically inactive women changed the mode to active after the surgery. Two women from the physically active group switched to the physically inactive group after the procedure. Due to such a small number, we did not perform statistical calculations for this change.

  • We have added a description of logistic regression:

“The influence of the domain scores on the physical activity was considered with binary logistic regression. Physical activity or lack thereof is a binary variable (active, inactive). In order to check whether a certain number of points awarded by patients after surgery within individual domains (domain scores) is related to undertaking physical activity, it was necessary to use this method. p-values ​​below 0.05 indicate a statistically significant relationship between undertaking physical activity depending on the level of quality of life assessment. The OR value, if the probability of taking up physical activity increases with an increase in the rating (within the domain) by one point, ie OR = 1.0403 means that an increase in domain scores by one point results in an increase in the probability of physical activity by 4%. "

Reviewer # 2:

Discussion

  • "The results of our research confirmed a significant correlation between the quality of life and the level of physical activity in the somatic and social spheres in female patients both before and after the SUI surgery with the use of TOT method." Unclear how authors came to this conclusion based on the presented data.
  • Extraneous information on general stress incontinence mechanisms is presented in the last 3 paragraphs of page 7.
  • Page 8 paragraph 1: Unclear as to how this study refutes the “hammock hypothesis” - the fact that patients are able to be physically active due to treated SUI suggests evidence to support the “hammock hypothesis” as this suggests the sling (which is thought to work based on the hammock hypothesis) is actually working.
  • The present data do not support conclusions 1-3.

Answer:

  • We changed the wording used to: " The results of our research confirm that, especially in physically active patients, the overall quality of life improved after the TOT procedure."
  • We removed this redundant information instead, using others that raised the level of the discussion.
  • Information regarding the hammock hypothesis was redundant in the discussion of this article and has been removed.
  • We improved conclusions 1-3:
  1. The TOT procedure increased physical activity in the majority of patients.
  2. After the surgery using the TOT method, improved the quality of life in the somatic and social domains in elderly women regarding the assessment of 12 months after the surgery.
  3. After the surgery using the TOT method, physically inactive women were able to become active and that active women were able to remain so, with attendant QoL gain

Round 2

Reviewer 1 Report

the authors have attempted to respond to the reviewer;s comments and have made clarifications - the paper does though need considerable editing for clarity  - much of the argument is there but it is often difficult to get at what the authors have found and mean  in their writing

Reviewer 2 Report

The authors have addressed all the requested modifications and editions.